# Portable Iontophoresis Device for Efficient Drug Delivery

**DOI:** 10.3390/bioengineering10010088

**Published:** 2023-01-09

**Authors:** Moonjeong Bok, Young Il Kwon, Zheng Min Huang, Eunju Lim

**Affiliations:** 1Department of Science Education/Creative Convergent Manufacturing Engineering, Dankook University, Yongin 16890, Republic of Korea; 2Nano-Convergence Mechanical Systems Research Division, Korea Institute of Machinery and Materials, Daejeon 34103, Republic of Korea; 3Department of Fiber System Engineering, Dankook University, Yongin 16890, Republic of Korea; 4Department of Convergence System Engineering, Dankook University, Yongin 16890, Republic of Korea

**Keywords:** iontophoresis, transdermal administration, diffusion, rhodamine, drug absorption

## Abstract

The timely delivery of drugs to specific locations in the body is imperative to ensure the efficacy of treatment. This study introduces a portable facial device that can deliver drugs efficiently using iontophoresis. Two types of power supplies—direct current and pulse ionization supplies—were manufactured by injection molding. Electrical stimulation elements, which contained Ag metal wires, were woven into facial mask packs. The diffusion phenomenon in the skin and iontophoresis were numerically modeled. Injection molding was simulated before the device was manufactured. Analysis using rhodamine B demonstrated a remarkable increase in the moisture content of the skin and effective absorption of the drug under an applied electric field upon the application of iontophoresis. The proposed concept and design constitute a new method of achieving effective drug absorption with wearable devices.

## 1. Introduction

Recently, the demand for innovative medical devices based on bioelectronics for enhanced performance and cost-effectiveness has increased [1,2,3]. In particular, transdermal drug delivery systems have received considerable attention [4,5,6,7]. To increase the driving force for drug delivery, several physical stimuli, such as chemical enhancers, microneedles, and electric currents, have been applied [8,9,10,11,12,13,14,15,16,17,18,19,20,21,22,23,24,25,26]. For instance, the iontophoresis process induced by an electric current repels ions and induces an electric field [14,15,16]. Iontophoretic delivery generally improves penetration deep into the stratum corneum, the outer skin layer [17,18,19]. That is to say, the electric current increases the drug permeability of the skin, which is required for effective drug delivery. In general, the direct current (DC) or alternating current (AC) power sources and metallic electrodes for iontophoresis devices are relatively heavy, bulky, and expensive [27,28,29]. These factors limit the usage of such devices in daily life. Therefore, the size, weight, and cost must be reduced for commercialization of these devices.

Facial masks are among the most common skincare products. They are generally affordable and relatively accessible. Thus far, considerable efforts have been expended to ensure that various functions can be served by facial masks, such as “skin whitening” and moisturizing. Yet, the applications for facial masks are limited because drugs cannot penetrate the human skin effectively due to the barrier lipids. Physical methods can be used to deliver drugs to the dermal layer effectively. Among the various methods, the approaches of forming microsized holes and applying electric fields are mainly used. For instance, various treatments using microneedles, such as microneedle therapy systems (MTSs), are currently being used for drug delivery in medical fields such as dermatology and plastic surgery.

In this study, a portable ion power supply that delivers drugs to the skin was designed and manufactured, and the MTS method and iontophoresis effects were compared and analyzed. The absorption behaviors of the drugs were analyzed using fluorescence images and rhodamine B. To overcome the limitations of the iontophoresis device, we developed a portable, flexible, and stable electric facial mask system and applied iontophoresis to analyze drug penetration into the skin tissue. With this device, improved skin penetration of ionic drugs can be realized through ionic repulsion effects. The developed electric facial mask was evaluated in terms of the skin moisture content and absorption range. Furthermore, the electrical field was simulated to identify the water absorption through the current flow and the distribution of potential difference in the skin. In addition, we implemented a prototype portable electric facial mask with a power supply supported by the ear and evaluated its drug delivery efficacy.

## 2. Materials and Methods

### 2.1. Sample Preparation

Porcine skin was obtained from a cadaver (thickness = approximately 5 mm) and was used to extract the nonfreezing Franz-cell membrane (FCM) (Medi Kinetics Micro-pig^®^, Pyeongtaek, Republic of Korea). All the skin samples were stored at 4 °C until the tests were performed. The tests were approved by the Institutional Animal Care and Use Committee.

For the tests, rhodamine B (0.05 g, Sigma-Aldrich 83689, molecular weight 479.01 g/mol, Sigma-Aldrich, St. Louis, MO, USA) was dissolved in 1 mL of deionized water to prepare a rhodamine B solution. Subsequently, 20 μL of the prepared rhodamine B solution was poured on the surface of a nonfreezing FCM with dimensions of 3 cm × 3 cm × 5 mm, and the surface was left undisturbed for 20 min. Then, the surface of the FCM was scratched using a commercially available microneedle cartridge (needle type: TRI-M 025, Doo-eul, Seoul, Republic of Korea). Following that, 20 microliters of the prepared rhodamine B solution was poured on the surface of the FCM, and the surface was left undisturbed for 20 min. Next, a galvanic ionic device (E.L.F, Seoul, Republic of Korea) was used to produce a fine current of 100 μA. Galvanic ion stimulation was performed for 20 min. The facial masks in the experiment were evaluated on the forearm. For this, an electro-assisted facial mask formed by placing a Ag yarn in the nonwoven fabric and a typical nonwoven facial mask that does not support the electrical system were used.

### 2.2. Device Analysis

The device for measuring electrical stimulation had an electrode/FCM/rhodamine B/(electric facial mask)/electrode structure (Appendix A). To create an environment for smooth current flow and practical device use, an FCM and facial mask were used between the devices. The simulation was developed in the AC/DC module of COMSOL Multiphysics and was performed to model the application of the drug delivery system on the skin. Fluorescence images were captured to observe the relative absorption intensity using Image J (National Institutes of Health, Bethesda, MD, USA). Moldex3D^®^ software (CoreTech System Co., Ltd., Republic of Korea) based on the finite element method (FEM) was used to simulate the injection molding process. The fluorescent dye, rhodamine B, penetrating the skin was traced before and after the simple diffusion, microneedle, and galvanic ionic device treatments in vitro. The exposed side of the skin was examined under a microscope to evaluate the penetration and depth. The porcine cadaver skin samples were treated with a cryoblock and were cut into 10 μm sections along the *z*-axis direction and frozen. The sections were visualized under a fluorescence microscope (OLYMPUS, CKX41) with 4× and 20× objective lenses. The results were expressed as the mean ± standard deviation. All statistical data were analyzed by performing the Student’s *t*-tests. In all cases, *p* ≤ 0.001 was considered significant. The moisture measurements were conducted with a skin moisture and oil content analyzer (Vcare SK-8, VCARE Co., Seoul, Republic of Korea).

### 2.3. Numerical Modeling

The injection molding processes were analyzed numerically using the commercial simulation software Moldex3D^®^ based on the FEM. The governing equations for the numerical model considered the conservation of mass, linear momentum, and energy equations as follows [30,31]:(1)$$\frac{d\rho }{dt}+\rho \left(\nabla \cdot \tilde{v}\right)=0,$$
(2)$$\rho \frac{d\tilde{v}}{dt}=-\nabla P+\nabla \cdot \tilde{\tau }+\rho \tilde{g},$$
(3)$$\rho C_{p}\frac{dT}{dt}=\beta T\frac{dP}{dt}+\eta {\dot{\gamma }}^{2}+\nabla \cdot \tilde{q},$$
where ρ is the density, $\tilde{v}$ is the velocity vector, P represents the pressure, $\tilde{\tau }$ is the stress tensor, $\tilde{g}$ is the gravity vector, $C_{p}$ is the specific heat at constant pressure, β is the thermal expansion coefficient, η is the generalized Newtonian viscosity, $\dot{\gamma }$ is the magnitude of the shear rate tensor, and $\tilde{q}$ is the heat flux vector.

The numerical simulation was performed using the steady-state AC/DC module of COMSOL Multiphysics. The electric field and current of the mask were also modeled. The model domain was developed, and boundary conditions were imposed. The electrical conductivities (σ_electrode_, σ_vitaminC_, and σ_skin_) and relative permittivities (ε_electrode_, ε_vitamine C_, and ε_skin_) of the materials were included in the simulation using their reference values. Assuming static stationary currents and fields, the electric field E must satisfy the following equations:(4)∇⋅J=0,
(5)J=σE,
(6)E=−∇V,
where J is the electrical current, σ is the electrical conductivity, E is the electric field, and V is the applied electrical potential.

## 3. Results and Discussion

For the mass-diffusion analysis, a rhodamine B solution was used. A needle cartridge was employed to puncture microholes in the skin (Figure 1b). The arrows in the figure indicate the direction of the ensuing scratch. The drug was expected to diffuse through the perforations and scratches. Moreover, when an electric field was applied, the drug was delivered transdermally due to ionic repulsion that arose between the drug and the electrode (Figure 1c).

Fluorescence images were captured with rhodamine B to visualize the drug penetration in the skin tissue. Figure 2 presents a comparison of the drug absorption outcomes in the pore area of the skin tissue. In addition, it shows the diffusion characteristics of the control group, scratches generated by micro-sized needles, and an electric field in the case of iontophoresis. The skin structure consists of the stratum corneum, viable epidermis, and dermis, as shown in Figure 2(a-i). To improve drug absorption, holes were generated by scratching the skin with a microneedle cartridge. Absorption of the rhodamine B solution was observed after application of electricity. The rhodamine B solution was applied to the skin surface and was observed after 20 min. The rhodamine B that was used has a molecular weight of 500 Da. The rhodamine B solution has a positive charge and causes ionic repulsion. The results (Figure 2(a-i,a-iv)) confirmed that the rhodamine B solution was absorbed only at the surface of the stratum corneum layer unless special conditions for the skin were considered. Thus, the results suggested that moieties cannot easily diffuse deep inside the skin layer through a simple concentration gradient method. In the case of scratches, the rhodamine B solution penetrated the skin tissue through the perforations and furrows. The rhodamine B that diffused into the skin layer was effectively absorbed by the scratched pores due to the induced hydrophilicity. When electricity was used, the results confirmed that a large quantity of rhodamine B penetrated the skin layer. It is clear that the application of electricity could enable the dye to penetrate deep into the stratum corneum. Figure 2b compares the relative fluorescence intensity and penetration depth. Figure 2(b-i) presents an optical image, Figure 2(b-ii) depicts an image stained with 4′6′-diamidino-2-phenylindole, and Figure 2(b-iii) provides a fluorescence image. The fluorescence intensity comparison confirmed that the absorption of the rhodamine B solution was increased by approximately 4.5 times due to the electrical effects. The penetration depth in the dermal layer was approximately 320 μm. Therefore, we confirmed that the solution penetrated deeper when the current flowed. Figure 2(b-iv,b-v) illustrate the relative mean fluorescence intensities and relative depths associated with the skin tissue cells in the high-resolution images in Figure 2(b-i–b-iii).

Experiments to study the effects of electricity suggested the development of the application displayed in Figure 3. The skin dries out and thickens with age. To care for the aging skin, drying must be prevented and the moisture content must be replenished. In this regard, lotions and creams are commonly used as emollients. As mentioned above, transdermal delivery by simple concentration gradients becomes difficult due to the low permeability of the stratum corneum of the outer skin layer. Therefore, as depicted in Figure 3, the electro-assisted facial mask was applied to the forearm, and the effect was confirmed by observing the absorption layer of the ionizing essence. Figure 3(a-i) provides an image of a typical nonwoven facial mask that does not support the electrical system, and Figure 3(a-ii) illustrates an electrode-based facial mask, in which current flows through the electrode system formed by placing a Ag yarn in the nonwoven fabric. In the experiment, the moisture content measurement location was first marked on the forearm, and its moisture content was measured before placing the facial mask on it. Ionizing essence and vitamin C were applied to each facial mask, which was coalesced to the skin for 20 min. Then, electricity was applied using ultrasonic microdermabrasion (Figure 3(a-ii)). After 20 min, the moisture was removed from the facial mask and skin surface, and the moisture content on the forearm was remeasured. The moisture content of the skin over time with and without iontophoresis is compared in Figure 3(a-iii). As a result of iontophoresis, the moisture content increased from approximately 47% to 55%. Figure 3b displays the fluorescence and optically merged images of the skin tissue (stratum corneum, viable epidermis, and dermis), to which an electrically assisted facial mask was applied. The actual electric facial mask contained negatively charged essence and vitamin C. Thus, when electricity was applied, the drug was absorbed into the dermis.

An earphone-like product was considered for containing the ion power supply. The shell was attached to a facial mask. As illustrated in Figure 4(a-i), the product consists of four parts: body (gray), switch (orange), conductor (yellow), and tip (green). Among them, the body (Figure 4(a-ii)) and the fabrication process were simulated based on injection molding. Through injection molding simulation, we found and optimized the process conditions necessary to manufacture the earphone-like product. One pin gate was selected, and the number of finite elements was approximately fifty thousand (Figure 4(a-iii)). Polypropylene (PP) (Lupol HI-4352L) from LG Chemical was employed. The material properties for the simulation were obtained from Moldex3D. In this study, only the filling analysis was performed to verify the formability of the product. The resin was processed at a melting temperature of 225 °C, a mold temperature of 65 °C, and an injection time of 1.2 s. The maximum sprue pressure developed from 0.1 s to 2 s was assessed to determine the optimum filling time. Figure 4b displays the sprue pressure as a function of the filling time. The minimum value occurs at a filling time of 1.2 s. When the filling time is shorter than the optimal time (1.2 s), the injection pressure must be adequately high to force the material into the mold cavity within a short time. In contrast, if the filling time is longer than the optimal time (1.2 s), as the polymer is injected into the cooling mold, it will result in a lower temperature and higher viscosity, requiring a higher injection pressure.

In the case of the optimal time (1.2 s), the sprue pressure gradually increased with respect to the filling time and yielded 3 changing points (inset of Figure 4b). At the first point, the sprue pressure abruptly increased when the runner was entirely filled, and the cavity began to fill. Subsequently, when the upper part of the product with complex shapes was completely filled, the sprue pressure considerably increased to approximately 8 MPa, and the bottom part began to be filled. Finally, when 98% of the cavity volume was filled, the v–p switch occurred, and the pressure marginally decreased from the highest value of 34.6 MPa. The v–p switch value represents the switch-over point from filling to packing. When filled, the v–p switch occurred, and the pressure subtly decreased from the highest value of 34.6 MPa.

Thus, the v–p switch value was set to 98% because the machine required a buffer to prevent damage. Figure 5a depicts the melt front advancement in the cavity with the exception of the runner part with respect to time. As shown in Figure 5(a-i), the time required to fill 25% of the total volume was 0.627 s, which accounted for approximately 50% of the total filling time (1.226 s). This could be attributed to the complex topology of the upper part of the product’s structure and the low initial sprue pressure. Over time, the resin flowed to the bar part of the product with a simpler shape, and the sprue pressure dramatically increased. Therefore, the time for 75% volume filling was 1.026 s (Figure 5(a-ii)), which is approximately 84% of the total filling time.

The pressure distribution in the cavity is displayed in Figure 5b. The pressure gradually decreased toward the bottom of the product. The pressure peaked at 30.525 MPa near the gate, which is marginally lower than the maximum sprue pressure. This is because pressure drops occurred when passing through the runner. The upper part of the product had a high-pressure value >25 MPa. Moreover, the bar part of the product with a simple shape yielded pressure values ranging from 10 to 20 MPa.

Figure 5c illustrates the center temperature distribution of the cavity. The center temperature is the melt temperature of the middle layer in the thickness direction at a given time. It indicates the thermal energy supply of the fresh hot melt. The temperature distribution in the cavity is mostly uniform and >200 °C.

This means that the product has good processability with minor deformation and shrinkage during the injection molding process. The highest temperature is 224.734 °C in the area near the gate and the bottom part of the product. The area with the lowest temperature value of 73.071 °C is also in the upper part of the product. Since that area is thinnest, the cooling process is performed there first and results in lower temperatures than in the other areas. Figure 5a,b present an actual image and the output voltage, respectively, of a portable ion device designed to facilitate the flow of microcurrents. The device was manufactured with the use of molding injection. This ion device is portable, and it can easily be placed near the face using earphones when a facial mask is applied. When using a drug-coated facial mask, the user must first attach the mask to the face. Then, the user plugs the portable ion device into the ear and applies electricity while attaching it to the silver yarn of the facial mask. Figure 6 shows a waveform image of the designed portable ion device. Because the DC can cause skin irritation when employed for a long time, an AC square wave is used for the actual device, as shown in Figure 6b. Figure 7a depicts a facial mask in which electrodes are embedded, and Figure 7b shows an electric facial mask manufactured by inserting a Ag yarn into nonwoven fabric. In Figure 7a, the device consists of (i) a Ag yarn electrode fabricated with alternating Ag and polymer cross-spinning, (ii) nonwoven felt coated with vitamin C and essence, and (iii) facial skin. Similar to the initial test in which we applied the electric facial mask, the results for which are presented in Figure 3 above, the electric facial mask was attached to the face, and the ionizing essence, vitamin C, and moisture analyzer (Vcare SK-8, VCARE) were applied to the skin. Then, the absorption of the ionizing essence and vitamin C by the skin was investigated. The results showed an effective improvement in the penetration depth of the facial skin. Through this study, it was possible to investigate the effect of an efficient transdermal drug delivery system applicable to wearable devices.

A system was constructed to supply electricity to the facial mask pack for effective drug delivery. Figure 8a provides a schematic of the electric facial mask. Figure 8b,c present the effects of drug delivery to the skin as obtained using COMSOL Multiphysics software. The images depict the model used to study the current flowing in the ion devices when vitamin C was applied to the skin. Negatively charged vitamin C was applied, and negatively charged ionized essence and vitamin C were applied to the actual mask pack [32,33]. Specifically, the images show the Ag electrode, vitamin C, and skin models designed to mimic the conditions based on which microcurrent flowed in the facial mask. The thickness of the skin was 1 mm; the electrode was inserted in the middle of the vitamin C over the skin, and the height was 0.08 mm.

Figure 8b,c present the distributions of the electric potential and normalized current density on the skin when a current of 100 μA was applied to the electrode, respectively. The electric field and current were concentrated around the electrode, which significantly affected the migration of vitamin C. That indicated that a concentrated electric field can facilitate drug absorption through the electric facial mask.

## 4. Conclusions

In this study, a portable ionic device was designed and fabricated for efficient transdermal drug delivery using electric fields. The iontophoresis and manufacturing process were numerically modeled to simulate the device. The mass diffusion in the skin was analyzed using fluorescence images obtained through experiments on porcine skin, and the effect of the applied electric field was evaluated. The effective penetration of the drug under an applied ionic current was observed. The drug penetrated 320 μm into the skin. Ion permeation could facilitate drug absorption and moisture content in the skin. The moisture content was improved by approximately 10% when electricity was supplied to the device. The current and potential fields were simulated to elucidate the ionic repulsive force in the skin. In addition, by mathematically analyzing the injection molding process, numerical values such as pressure, temperature, and charging time, which were optimized, were applied to fabricate a portable electric-charge application device. The uniform temperature distribution led to relatively little deformation and shrinkage. This research provides authentic insights into the design and manufacture of a portable ionic medical device for implementation and commercialization.

## Figures and Tables

**Figure 1 bioengineering-10-00088-f001:**
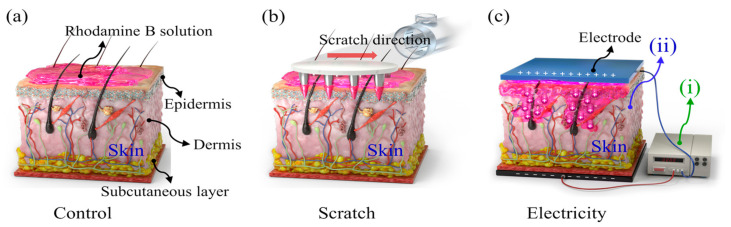
Schematics of (**a**) control, (**b**) needle scratch condition, and (**c**) electrically induced iontophoresis for transdermal transmission. Schematics of (**i**) power supply and (**ii**) skin.

**Figure 2 bioengineering-10-00088-f002:**
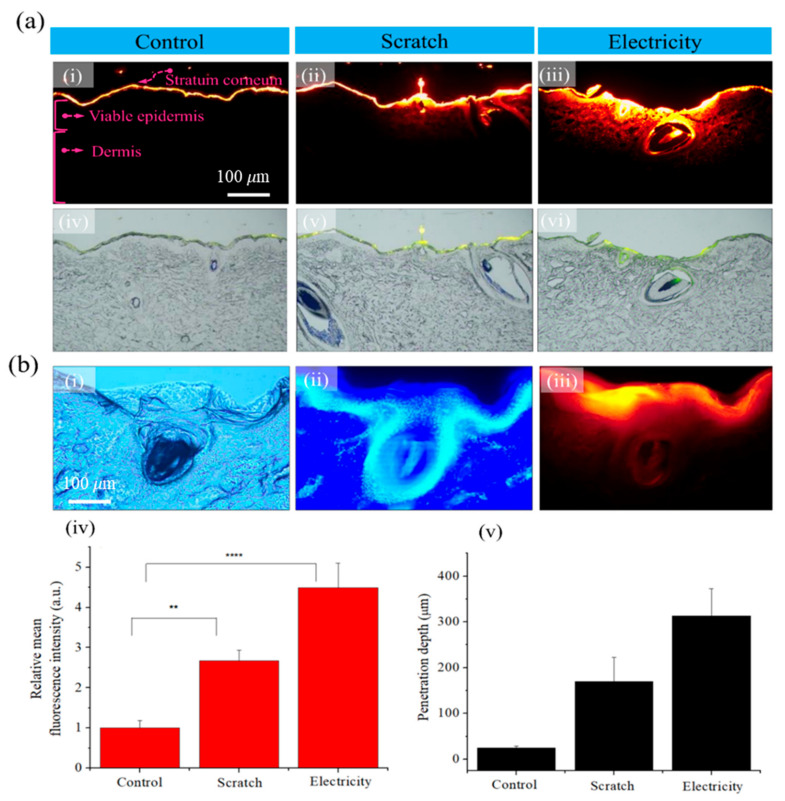
(**a**) Fluorescence and merged optical images. (**i**,**iv**) Control, (**ii**,**v**) scratch effects, and (**iii**,**vi**) electricity effects. (**b**) Magnified images obtained when electricity was applied. (**i**) Bright field image, (**ii**) 4′6′-diamidino-2-phenylindole staining image, (**iii**) fluorescence image, (**iv**) fluorescence distribution using Image J (here, ** indicates *p* < 0.05 and **** indicates *p* < 0.01), and (**v**) penetration depth (the scale bars in (**a-i**)–(**a-vi**) = 100 μm and in (**b-i**)–(**b-iii**) = 100 μm).

**Figure 3 bioengineering-10-00088-f003:**
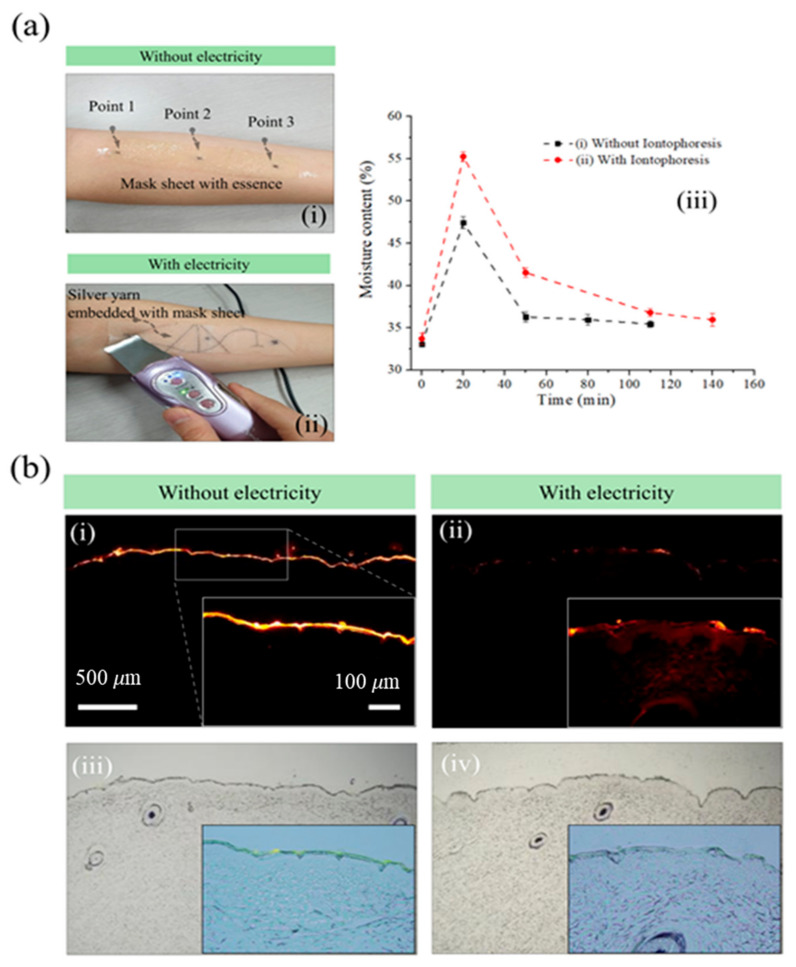
Iontophoresis effects. (**a**) Actual experimental images (**i**) without electricity and (**ii**) with electricity, and (**iii**) comparison of moisture content (%) with and without electricity. (**b**) Fluorescence and merged optical images with and without electricity for electrically assisted facial mask. (**i**,**iii**) Without electricity, and (**ii**,**iv**) with electricity (scale bars: 500 μm, 100 μm).

**Figure 4 bioengineering-10-00088-f004:**
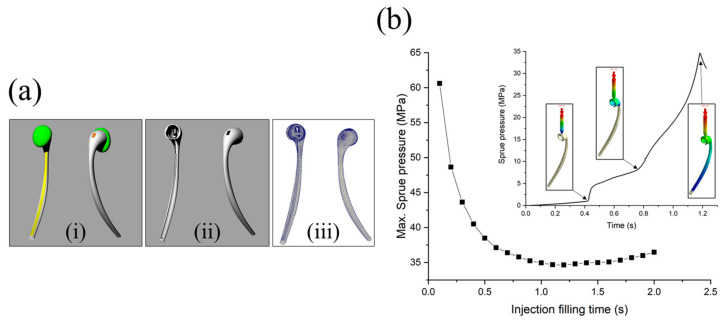
(**a-i**) Geometrical structure of the earphone-like product. (**a-ii**) Computer-aided design model and (**a-iii**) mesh generation for the injection part. (**b**) Maximum sprue pressure with respect to filling time. The inset shows the variation of the sprue pressure at a filling time of 1.2 s.

**Figure 5 bioengineering-10-00088-f005:**
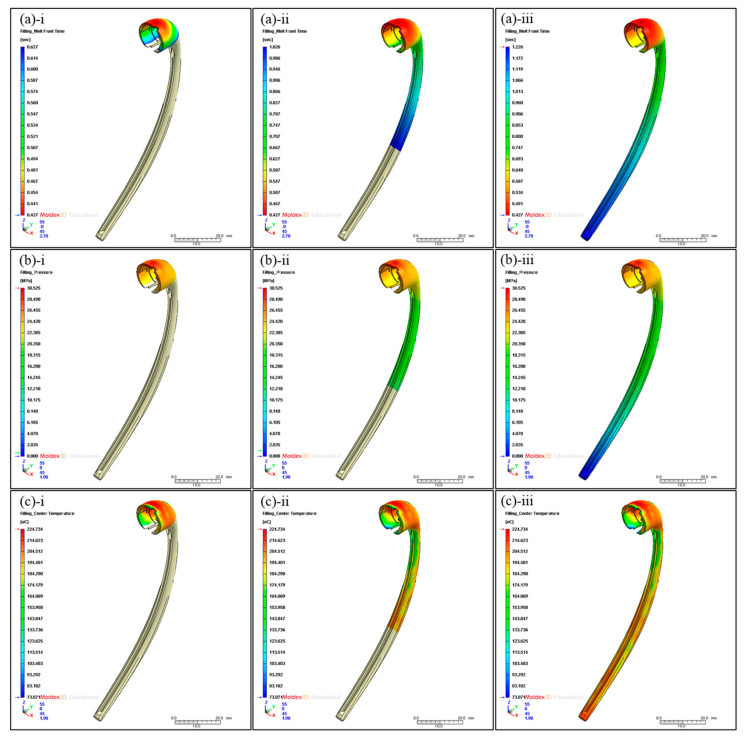
(**a**) Melt front of polymer in the mold cavity with respect to time: (**a-i**) 25%, (**a-ii**) 75%, and (**a-iii**) 100% filling. (**b**) Pressure distribution in the mold cavity with respect to time: (**b-i**) 25%, (**b-ii**) 75%, and (**b-iii**) 100% filling. (**c**) Center temperature distribution in the mold cavity with respect to time: (**c-i**) 25%, (**c-ii**) 75%, and (**c-iii**) 100% filling.

**Figure 6 bioengineering-10-00088-f006:**
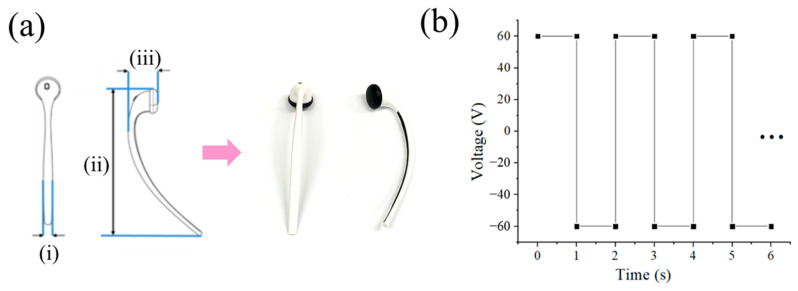
Galvanic ionic device for current flow. (**a**) Designed ionic device at different scales: (**i**) length: 6.00, (**ii**) length: 99.50, and (**iii**) length: 20.10. The arrow indicates the actual manufactured product image. (**b**) Continuous waveform from the real designed portable ionic device.

**Figure 7 bioengineering-10-00088-f007:**
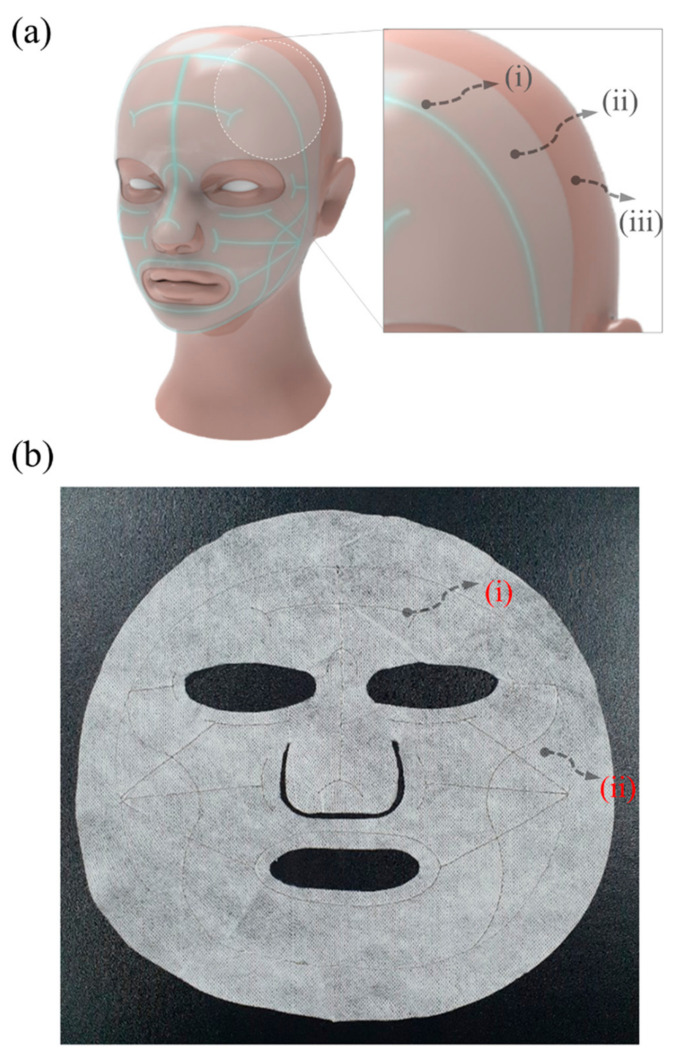
(**a**) Application of facial mask with an embedded electrode, and (**b**) produced electric facial mask. The arrows in the enlarged part indicate the (**i**) Ag yarn, (**ii**) nonwoven felt, and (**iii**) facial skin.

**Figure 8 bioengineering-10-00088-f008:**
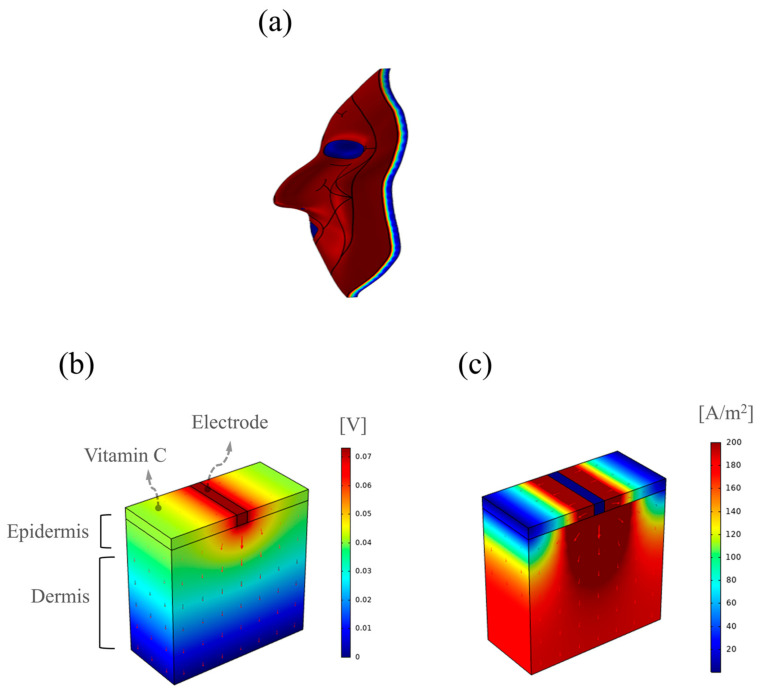
Numerical simulation of the application of the electrically assisted facial mask. (**a**) Expected drug delivery on facial mask skin, (**b**) electrical potential (red arrows: direction of electric field), and (**c**) normalized current density (red arrows: direction of current density).

## Data Availability

Data are contained within the article and Appendix A.

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
