# Peer review of "Portable Iontophoresis Device for Efficient Drug Delivery"

_bioengineering, 2023, doi:10.3390/bioengineering10010088_

Round 1

Reviewer 1 Report

This work presented an efficient way of drug delivery to facial skin. However, there are doubts that this device has real applications. 

1. the simulation on the ear-like device seemed irrelevant to the performance of the device.

2. the details of the device, such as working electrical current or voltage is not given.

Additional comments:

3. What specific improvements should the authors consider regarding the
methodology? What further controls should be considered?
The details of operation by this device should be provided, such as the current that patient experiences, etc for safety concern.

4. Please include any additional comments on the tables and figures.
Figure 6 is irrelevant.

Author Response

We thank the reviewer for the helpful comments. Please see the attachment.

Reviewer 2 Report

In this paper, the authors developed a portable ionic device for trans-dermal drug delivery using electric fields, which was both confirmed by Simulations and experimental tests. Generally, this paper is clearly organized, while, there are still some issues to be addressed before final decision:

1, the research background should be further broadened by discussing related references, such as microelectrodes (https://doi.org/10.1016/j.bios.2020.112413; https://doi.org/10.1016/j.biomaterials.2017.03.019),  even electrodynamic approaches (https://doi.org/10.1002/advs.202001223; https://doi.org/10.1021/acsami.1c19985; https://doi.org/10.1021/acsnano.1c08544), to further exhibit the advantages of electric field-based method.

2, The problem solved by this work, should be able to be solved by microneedle array (https://doi.org/10.1016/j.ejpb.2020.12.006), so what is the advantage of this work? In the meanwhile,  this method requires the drugs to be charged, which should also be discussed.

3, in this case, the mechanism for the drug accumulation should be further explained, in the experimental situation, why Rodanmine B is positively charged? If applied in vivo, will the conditions the same? 

4, the paper could be more concise.

5, the language could be further polished.

Author Response

(The authors gave the same response as above.)

Reviewer 3 Report

In its current form, the article is not acceptable. Section Materials and Methods is written very poorly. This chapter is very chaotic and there is a lot of information missing. Therefore, it is hard to understand text in the Results and discussion section. How moisture was assessed? Why vitamin c was used? Is the portable ionic device designed to work with the facial mask? How they can be assembled? Why the facial mask was not tested on face? Results and Discussion section has not a logical order of chapters. There is no any real discussion of the results. In my opinion this article must be written once again more carefully.

Author Response

(The authors gave the same response as above.)

Round 2

Reviewer 1 Report

1. The modelled drug diffusion depth is relatively shallow, the authors should justify the efficiency of drug delivered to the targets.

2. The scale bar should be given in Fig3(b).

Author Response

(The authors gave the same response as above.)

Reviewer 2 Report

Most of the questions have been addressed by the authors, leading to a clear improvement of the paper quality, thus could be considered for publication.

Author Response

We thank the reviewer for the helpful comments.

Reviewer 3 Report

- Assembly method of the mask and ear-type device must be explained in details or some scheme must be included.

- experiment reported in figures 1 and 2 should include an additional control group: applying electrical field without scratching.

- Evaluation of the mask on the forearm should be described in the Materials and Methods section.

- the publication practically does not contain an element of discussion of the results in the light of other studies

Author Response

  1. Assembly method of the mask and ear-type device must be explained in details or some scheme must be included.

We thank the reviewer for the helpful comments. When using a drug-coated facial mask, attach the facial mask to the face first. Then, plug the portable ion device into the user's ear and apply electricity while attaching it to the silver yarn embedded in the facial mask. We added above comments in L 247-249.

  1. Experiment reported in figures 1 and 2 should include an additional control group: applying electrical field without scratching.

We thank the reviewer for the helpful comments. In the case of electricity effect in this study, electrical field was applied without scratching. In other words, the three cases in the experiment are normal conditions, scratch conditions, and electricity conditions. In this study, we did not proceed with the case of applying electricity after scratching (considering both scratch effect and electricity effect). We will proceed with a more detailed step as you mentioned in our further study

  1. Evaluation of the mask on the forearm should be described in the Materials and Methods section.

We thank the reviewer for the helpful comments. The facial mask used in the experiment was evaluated for the forearm with an electro-assisted facial mask formed by placing an Ag yarn in the non-woven fabric and a typical non-woven facial mask that does not support the electrical system. we added above comments in L 74-76.

  1. The publication practically does not contain an element of discussion of the results in the light of other studies

We thank the reviewer for the helpful comments. There are many studies on drug delivery using the iontophoresis method. We added above comments and references for the iontophoretic drug delivery in L 29-31.

Round 3

Reviewer 3 Report

The sentence " When electricity was used in the area with pores, the results confirmed that a large amount of rhodamine B penetrated the skin layer." (L147-148) is misleading as it suggest, that elictricity was applied in combination with scratching. It should be corrected and experimental groups should be more clearly desribed.

Author Response

(The authors gave the same response as above.)
